# Optimization and Antibacterial Evaluation of Novel 3-(5-Fluoropyridine-3-yl)-2-oxazolidinone Derivatives Containing a Pyrimidine Substituted Piperazine

**DOI:** 10.3390/molecules28114267

**Published:** 2023-05-23

**Authors:** Xin Wang, Bo Jin, Yutong Han, Tong Wang, Zunlai Sheng, Ye Tao, Hongliang Yang

**Affiliations:** 1Department of Veterinary Medicine, Northeast Agricultural University, Harbin 150030, China; 2Heilongjiang Key Laboratory for Animal Disease Control and Pharmaceutical Development, Harbin 150030, China

**Keywords:** 3-(5-fluoropyridine-3-yl)-2-oxazolidinone derivatives, antibacterial activity, molecular docking, antibiofilm activity, drug resistance

## Abstract

In this study, a series of novel 3-(5-fluoropyridine-3-yl)-2-oxazolidinone derivatives were designed and synthesized based on compounds previously reported, and their antibacterial activity was investigated. Then their antibacterial activity was investigated for the first time. Preliminary screening results showed that all these compounds exhibited antibacterial activity against gram-positive bacteria, including 7 drug-sensitive strains and 4 drug-resistant strains, among which compound **7j** exhibited an 8-fold stronger inhibitory effect than linezolid, with a minimum inhibitory concentration (MIC) value of 0.25 µg/mL. Further molecular docking studies predicted the possible binding mode between active compound **7j** and the target. Interestingly, these compounds could not only hamper the formation of biofilms, but also have better safety, as confirmed by cytotoxicity experiments. All these results indicate that these 3-(5-fluoropyridine-3-yl)-2-oxazolidinone derivatives have the potential to be developed into novel candidates for the treatment of gram-positive bacterial infections.

## 1. Introduction

Bacterial infections can lead to skin suppuration, bacteremia, local and systemic inflammation, and other serious diseases [1]. With the discovery and development of antibiotics, revolutionary changes have taken place in the treatment of bacterial infections, effectively reducing infection rates and mortality. However, the irregular use of antimicrobial drugs has led to the emergence of various drug-resistant strains, which are considered dangerous and stubborn clinical pathogens that cause difficult-to-treat, life-threatening illnesses [2]. Furthermore, it has been found in clinical studies that many pathogenic bacteria can adhere to the surfaces of objects, secrete metabolites, and generate extracellular polymeric substances (EPS), thus forming biofilms with a “cell population-metabolite” structure, which can effectively protect strains and produce drug resistance, leading to a high incidence of nosocomial infection and a high treatment cost [3,4]. Although great efforts have been made to create novel and effective antimicrobial methods, the pace of development is too slow to meet clinical needs [5,6,7,8].

At present, many pharmaceutical chemists are trying to develop new antibacterial drugs with novel structures, unique mechanisms of action, and long-term effectiveness [9,10,11]. Oxazolidinones are a class of chemosynthetic antibacterial drugs with a brand-new chemical structure, similar to sulfonamides and quinolones, which are used to treat skin and tissue infections, pneumonia, untreatable bacterial infections, and other infectious diseases caused by gram-positive bacteria [12]. Oxazolidinones have been widely used and studied because of their unique antibacterial mechanism, that is, inhibition of protein synthesis at the initial stage and no cross-resistance with other antibacterial drugs [13,14,15,16]. Moreover, some oxazolidinone analogues have been marketed previously [17,18,19]. Although the initial effect is satisfactory, drug resistance appears after long-term use, accompanied by thrombocytopenia and other adverse reactions [20]. This has prompted pharmaceutical chemists to continue to improve oxazolidinone antimicrobials with higher antimicrobial activity and sustained sensitivity [21,22,23,24,25,26,27,28,29].

Our research group previously modified the structure of linezolid (Figure 1) [30,31,32,33,34], found that compounds **1** and **2** have efficient antibacterial activity for *S. aureus*, *Streptococcus pneumoniae*, *Enterococcus faecalis,* etc. (MICs = 4~64 µg/mL) and exciting antibiofilm activity (MBIC = 0.5~8 µg/mL). The vinyl structure in these molecules is retained and cyclized to form a new aromatic heterocyclic ring. Considering the unique electronegativity of pyrimidines and their ability to form hydrogen bonds [35], pyrimidine aromatic rings were introduced into the structure [36]. Accordingly, a series of 3-(5-fluoropyridine-3-yl)-2-oxazolidinone derivatives were synthesized and tested for antibacterial activity (Figure 2).

## 2. Results and Discussion

### 2.1. Chemistry

According to the steps described in the literature [37], intermediate **4** was produced through a multi-step reaction by commercially purchased compound **3**. After removing the Boc protection group, compound **4** was coupled with 2,4-dichloropyrimidine to produce intermediate **5**. The intermediate **5** was linked with different amines to generate the final products **6a-m**, as displayed in Figure 1. Furthermore, the final products **7a-n** were obtained by a similar method.

### 2.2. Antibacterial Activity Assay

#### 2.2.1. Minimum Inhibitory Concentration against Standard Strains 

Intermediate **5** and further derivatives **6a-m** were synthesized and tested against a panel of gram-positive bacteria using the double dilution method. As shown in Table 1, all these compounds had moderate antibacterial activity against all six tested gram-positive bacteria but no activity against gram-negative bacteria (*E. coli*). The MICs of compounds **6a-m** were 2~32 µg/mL against gram-positive bacteria, while the MIC was 1~2 µg/mL when R^1^ = Cl (**5**), which had better antibacterial activity. It was speculated that the cavity of the target near the R^1^ side chain was not large enough for these substituent groups.

After that, a series of pyrimidine derivatives, **7a**-**n** without a further substituted group, was synthesized, and their antibacterial activity was tested (Table 2). All these compounds, except **7m**, had better antibacterial activity than compounds **1** and **2**, but these compounds still exhibited no effect against gram-negative bacteria. Among them, compound **7j** exhibited the best activity with a MIC of 0.25~1 µg/mL. First, while keeping X^1^ = N, three substituent groups on the pyrimidine ring (R^1^ = Cl, NH_2_, or R^1^ = R^3^ = Cl) were examined (**5**, **7e** and **7h**). All of them exhibited better activities than the compounds with X^1^ = C (**7b**, **7f** and **7g**). Then, by comparing the activity of chlorine-substituted compounds **5**, **7a**, **7h**, **7j** and **7n**, it was found that the number of chlorine atoms had no significant influence on activity. Meanwhile, keeping X^1^, X^2^, R^1^, and R^3^ constant (X^1^ = N, X^2^ = C, R^1^ = Cl, and R^3^ = H), F, Cl, Br, and methyl substituent groups on R^4^ were examined. All these derivatives displayed similar biological activities (MICs = 0.25~4 µg/mL), indicating that F, Cl, Br, or methyl were acceptable as substituents.

#### 2.2.2. Minimum Inhibitory Concentration against Drug-Resistant Strains

After evaluating the antibacterial potential of these derivatives, they were further tested against clinically isolated resistant bacteria. As shown in Table 3, these MIC results show that compounds **7i-l** had significant antibacterial activity against MRSA and VRE but no effect against linezolid-resistant strains.

### 2.3. Molecular Docking Study

To understand binding site, state, conformation, and interaction, the promising compound **7j** was selected for further docking study with the 50S ribosomal subunit from *Haloarcula Marismortui* (PDB ID: 3CPW) [38,39]. As shown in Figure 2, the compound that expanded linearly bound to the peptidyl transferase center (PTC) of the 50S ribosomal subunit. The potential compound existed in the cavity of PTC, which was composed by U2619, U2540, G2539, U2583, U2538, C2486, G2101, and A2485. Moreover, the H atom and O atom on the 5-side chain amide group of the oxazolidinone ring formed a hydrogen bond with A2485 and A2636, respectively. In addition, the pyrimidine ring of compound **7j** and the pyrimidine ring formed **π-π** conjugations with C2486 and U2538.

As can be seen from Figure 3, the 2-Cl atom on the pyrimidine substituent extended into a shallow pocket, which was too small to accommodate other groups on the pyrimidine. That was probably the reason why compounds **6a-m** had worse antibacterial activity. Meanwhile, there was a larger space between the 5-Cl atom on the pyrimidine substituent and the surface of the cavity, which might be the reason why compounds **7i-m** had a better antibacterial effect. 

### 2.4. Inhibition of Biofilm Formation

Using the microtiter dish biofilm formation assay [40], four potent compounds were selected for further evaluation of their effects on bacterial biofilm formation against four drug-resistant strains. As shown in Table 4, the results show that all these compounds significantly inhibited the formation of biofilms, with the minimum biofilm inhibitory concentrations (MBICs) of 0.5 µg/mL against MRSA and VRE and 1~4 µg/mL against LRSA and LRSP. The above results indicate that these compounds can significantly inhibit the growth of biofilm, and it could be speculated that they have stable effects and do not easily develop resistance to bacteria. Meanwhile, the results show that all compounds are more effective than linezolid against four drug-resistant strains, which indicates all compounds have different mechanisms with linezolid.

### 2.5. Cytotoxicity Determination

When a chemical substance is used to treat infection, it may affect the physiological activity of both cells and bacteria, thereby reducing the cell’s survival rate [41]. Therefore, it was necessary to evaluate the toxicity of active derivatives. The cytotoxicity of compound **7j** against the Hela cell line was detected via the MTT colorimetric assay, as shown in Table 5. The result shows that the cytotoxicity of the tested compound increased in a dose-dependent manner, and cell survival at 256 µg/mL and lower concentrations was higher than 85%. Considering that the cytotoxicity only appeared above 256 µg/mL, which was 64~512 times that of its MICs. Consequently, the compound **7j** has the potential to be further developed as an antibacterial drug.

## 3. Experimental Section

### 3.1. Materials and Methods

All the chemicals and solvents used in this study were of analytical grade. All the reagents were purchased from Tianjin Tianli Chemical Reagent Co., Ltd., Tianjin, China. All solvents and chemicals were purified by standard methods. Unless otherwise stated, the synthesis of all compounds was monitored by thin layer chromatography (TLC) and purified by rapid column chromatography. Thin layer chromatography (TLC) was performed on silica gel G plates (Taizhou Luqiao Sijia Biochemical Plastic Products Factory, Taizhou, China). The melting point (m. p.) of all products was measured by the SGW X-4A micro-melting point meter apparatus. All compounds were tested to verify their purity by HPLC (Shimadzu Corporation, Kyoto, Japan). Using the Diamonsil C18 column, the mobile phase was acetonitrile and water of different gradients at a flow rate of 1.0 mL•min^−1^. The column temperature was set at 35 °C and the injection volume of each sample was 10 μL. Every sample was quantitatively diluted with methanol to 1 mg•mL^−1^. ^1^H NMR spectra (600 MHz) and ^13^C NMR spectra (150 MHz) were recorded on a Bruker Advance spectrometer with tetramethyl-silane (TMS) as the internal standard and DMSO-d_6_ as the solvent. The used standard strains were purchased from the American Type Culture Collection (ATCC), and the drug-resistant strains were isolated from clinical sources. The Hela cells were donated by Dr. Tao Wang (Qiqihar Medical University). 

### 3.2. Chemistry

#### 3.2.1. Synthesis of (S)-N-((3-(6-(4-(2-chloropyrimidin-4-yl)piperazin-1-yl)-5-fluoropyridin-3-yl)-2-oxooxazolidin-5-yl)methyl)acetamide(5)and(S)-N-((3-(6-(4-(4-chloropyrimidin-2-yl)piperazin-1-yl)-5-fluoropyridin-3-yl)-2-oxooxazolidin-5-yl)methyl)acetamide (**7b**)

A solution of compound **4** (218 mg, 0.5 mmol) in DCM (5 mL) at 0 °C was dropwise added to TFA (1 mL) and then stirred for 2 h. After the reaction was complete, TEA was added to the solution at 0 °C to adjust pH. The filtrate was concentrated in vacuo. To a solution of the concentrate in ethanol (3 mL) was added TEA (0.14 mL, 1 mmol) and 2, 4-dichloropyrimidine (97 mg, 0.65 mmol), and then stirred at reflux overnight. After the reaction was complete and concentrated, the mixture was extracted with DCM (5 mL × 3). The organic phase was washed with brine and concentrated in vacuo. The residue was purified by silica gel column chromatography (DCM/MeOH/TEA = 50:1:1) to yield compounds **5** and **7b**. 

Compound **5** was white solid; yield 5.8%. m. p. 171.1–173.1 °C. ^1^H NMR (600 MHz, DMSO-d_6_) δ 8.32 (t, J = 6.0 Hz, 1H), 8.14 (d, J = 2.4 Hz, 1H), 8.11 (d, J = 6.0 Hz, 1H), 7.94 (dd, J = 14.4, 2.4 Hz, 1H), 6.88 (d, J = 6.0 Hz, 1H), 4.79 − 4.72 (m, 1H), 4.13 − 4.10 (m, 1H), 3.79 − 3.72 (m, 5H), 3.46 − 3.40 (m, 6H), 1.84 (s, 3H). ^13^C NMR (150 MHz, DMSO-d_6_) δ 170.6, 163.0, 160.0, 158.0, 154.8, 149.1 (d, JC-F = 257.1 Hz), 145.6, 133.0, 130.0, 115.4, 102.9, 72.6, 55.4, 47.6, 47.4, 41.9, 22.9. HRMS (ESI) (positive mode) *m*/*z* calculated for C_19_H_21_ClFN_7_O_3_: 449.87; Found: 450.129.

Compound **7b** was white solid; yield 40.4%. m. p. 160.0–161.1 °C. 1H NMR (600 MHz, DMSO-d6) δ 8.35 (d, J = 5.4 Hz, 1H), 8.25 (t, J = 6.0 Hz, 1H), 8.13 (d, J = 2.4 Hz, 1H), 7.93 (dd, J = 14.4, 2.4 Hz, 1H), 6.77 (d, J = 5.4 Hz, 1H), 4.89 − 4.63 (m, 1H), 4.21 − 4.02 (m, 1H), 3.82 − 3.79 (m, 4H), 3.75 − 3.67 (m, 1H), 3.44–3.33 (m, 5H), 1.84 (s, 3H). ^13^C NMR (150 MHz, DMSO-d_6_) δ 170.5, 161.5, 160.5, 154.8, 149.1 (d, JC-F = 257.1 Hz), 145.9, 133.0, 130.0, 128.7, 115.4, 109.7, 72.6, 47.7, 46.1, 43.7, 41.9, 22.9. HRMS (ESI) (positive mode) *m*/*z* calculated for C_19_H_21_ClFN_7_O_3_: 449.87; Found: 450.119.

Raw data for the above products are presented in Appendix A.

#### 3.2.2. General Procedure for the Synthesis of **6a** and **6b**

A solution of compound **5** (200 mg, 0.44 mmol) in an amine solution (4 mL) was stirred at room temperature for three days. After the reaction was complete, the filtrate was concentrated in vacuo. The mixture was extracted with DCM (5 mL × 3). The organic phase was washed with brine and concentrated in vacuo. The residue was purified by silica gel column chromatography (DCM/MeOH = 30:1) to yield compounds **6a** and **6b**.

*(S)-N-((3-(5-fluoro-6-(4-(2-(methylamino)pyrimidin-4-yl)piperazin-1-yl)pyridin-3-yl)-2-oxooxazolidin-5-yl)methyl)acetamide (**6a**)*.

Compound **6a** was a yellow solid; yield 40.5%. m. p. 193.3–195.1 °C. ^1^H NMR (600 MHz, DMSO-d_6_) δ 8.27 (t, J = 6.0 Hz, 1H), 8.14 (d, J = 2.4 Hz, 1H), 7.94 (dd, J = 14.4, 2.4 Hz, 1H), 7.86 (d, J = 6.6 Hz, 1H), 7.31 − 7.30 (s, 1H), 6.31 (d, J = 6.6 Hz, 1H), 4.78 − 4.73 (m, 1H), 4.13 − 4.10 (m, 1H), 3.84 − 3.78 (s, 4H), 3.75 − 3.73 (m, 1H), 3.43 − 3.42 (m, 6H), 2.82 (d, J = 4.8 Hz, 3H), 1.84 (s, 3H).^13^C NMR (150 MHz, DMSO-d_6_) δ 170.5, 162.0, 154.8, 149.1 (d, JC-F = 257.1 Hz), 145.7, 145.7, 133.0, 133.0, 130.0, 115.6, 115.5, 72.6, 47.7, 47.6, 43.9, 41.9, 28.1, 22.9. HRMS (ESI) (positive mode) *m*/*z* calculated for C_20_H_25_FN_8_O_3_: 444.47; Found: 445.150.

*(S)-N-((3-(5-fluoro-6-(4-(2-(isopropylamino)pyrimidin-4-yl)piperazin-1-yl)pyridin-3-yl)-2-oxooxazolidin-5-yl)methyl)acetamide* (**6b**).

Compound **6b** was a pink solid; yield 44.4%.m. p. 196.8–199.5 °C.^1^H NMR (600 MHz, DMSO-d_6_) δ 8.24 (t, J = 6.0 Hz, 1H), 8.13 (d, J = 2.4 Hz, 1H), 7.93 (dd, J = 14.4, 2.4 Hz, 1H), 7.82 (d, J = 6.0 Hz, 1H), 6.31 (s, 1H), 6.04 (d, J = 6.0 Hz, 1H), 4.79 − 4.72 (m, 1H), 4.12 − 4.10 (m, 1H), 4.04 − 3.95 (m, 1H), 3.74 − 3.72 (m, 1H), 3.68 − 3.66 (m, 4H), 3.49 − 3.36 (m, 5H), 1.84 (s, 3H), 1.12 (d, J = 6.6 Hz, 6H), 1.08 − 0.94 (m, 1H). ^13^C NMR (150 MHz, DMSO-d_6_) δ 170.5, 162.7, 161.7, 157.2, 154.8, 149.1 (d, JC-F = 256.3 Hz), 146.0, 133.0, 129.9, 115.5, 115.4, 72.6, 47.8, 47.7, 47.6, 43.5, 42.2, 41.9, 23.1, 22.9. HRMS (ESI) (positive mode) *m*/*z* calculated for C_22_H_29_FN_8_O_3_: 472.53; Found: 473.189.

Raw data for the above products are presented in Appendix A.

#### 3.2.3. General Procedure for the Synthesis of **6c–m**

To a solution of compound **5** (150 mg, 0.33 mmol) in dioxane (4 mL), p-toluene sulfonic acid monohydrate (11.4 mg, 0.066 mmol) and amine (1 mmol) were added and stirred at reflux overnight. After the reaction was complete, the filtrate was concentrated in vacuo. The mixture was extracted with DCM (5 mL × 3). The organic phase was washed with brine and concentrated in vacuo. The residue was purified by silica gel column chromatography (DCM/MeOH = 30:1) to yield compounds **6c–m.**

*(S)-N-((3-(6-(4-(2-((2,2-difluoroethyl)amino)pyrimidin-4-yl)piperazin-1-yl)-5-fluoropyridin-3-yl)-2-oxooxazolidin-5-yl)methyl)acetamide* (**6c**).

Compound **6c** was a pink solid; yield 66.7%.m. p. 228.1–231.2 °C.^1^H NMR (600 MHz, DMSO-d_6_) δ 8.24 (t, J = 6.0 Hz, 1H), 8.13 (d, J = 2.4 Hz, 1H), 7.94 (dd, J = 14.4, 2.4 Hz, 1H), 7.87 (d, J = 6.0 Hz, 1H), 7.02 (s, 1H), 6.20 (d, J = 6.0 Hz, 1H), 4.79 − 4.72 (m, 1H), 4.13 − 4.10 (m, 1H), 3.76 − 3.73 (m, 1H), 3.73 − 3.68 (m, 5H), 3.68 − 3.58 (m, 2H), 3.46 − 3.38 (m, 6H), 1.84 (s, 3H). ^13^C NMR (150 MHz, DMSO-d_6_) δ 170.5, 162.5, 154.8, 149.1 (d, JC-F = 257.2 Hz), 145.9, 145.8, 133.1, 133.0, 129.9, 115.6, 115.4, 72.6, 47.7, 47.64, 43.9, 43.7, 43.6, 41.9, 22.9. HRMS (ESI) (positive mode) *m*/*z* calculated for C_21_H_25_F_3_N_8_O_3_: 494.48; Found: 495.163.

*(S)-N-((3-(6-(4-(2-(allylamino)pyrimidin-4-yl)piperazin-1-yl)-5-fluoropyridin-3-yl)-2-oxooxazolidin-5-yl)methyl)acetamide* (**6d**) 

Compound **6d** was a white solid; yield 68.1%.m. p. 182.1–184.6 °C.^1^H NMR (600 MHz, DMSO-d_6_) δ 8.25 (t, J = 6.0 Hz, 1H), 8.14 (d, J = 2.4 Hz, 1H), 7.94 (dd, J = 14.4, 2.4 Hz, 1H), 7.85 (d, J = 6.6 Hz, 1H), 7.51 − 7.08 (m, 1H), 6.28 (d, J = 6.0 Hz, 1H), 5.92 − 5.86 (m, 1H), 5.19 (d, J = 17.4 Hz, 1H), 5.08 (d, J = 10.2 Hz, 1H), 4.77 − 4.73 (m, 1H), 4.13 − 4.10 (m, 1H), 3.93 − 3.91 (m, 2H), 3.78 − 3.75 (m, 4H), 3.74 − 3.72 (m, 1H), 3.43 − 3.39 (m, 6H), 1.84 (s, 3H). ^13^C NMR (150 MHz, DMSO-d_6_) δ 170.5, 162.0, 154.8, 149.1 (d, JC-F = 257.5 Hz), 145.8, 138.1, 136.4, 133.0, 130.0, 128.5, 126.0, 115.6, 115.4, 72.6, 47.6, 43.9, 43.4, 41.9, 22.9, 21.2. HRMS (ESI) (positive mode) *m*/*z* calculated for C_22_H_27_FN_8_O_3_: 470.51; Found: 471.133.

*(S)-N-((3-(5-fluoro-6-(4-(2-(prop-2-yn-1-ylamino)pyrimidin-4-yl)piperazin-1-yl)pyridin-3-yl)-2-oxooxazolidin-5-yl)methyl)acetamide* (**6e**)

Compound **6e** was a white solid; yield 44.9%. m. p. 185.6–188.7 °C.^1^H NMR (600 MHz, DMSO-d_6_) δ 8.29 (t, J = 6.0 Hz, 1H), 8.14 (d, J = 2.4 Hz, 1H), 7.94 (dd, J = 14.4, 2.4 Hz, 1H), 7.88 (d, J = 6.0 Hz, 1H), 7.22 (s, 1H), 6.23 (d, J = 6.0 Hz, 1H), 5.33 (s, 1H), 4.77 − 4.73 (m, 1H), 4.13 − 4.10 (m, 1H), 4.04 − 4.03 (m, 2H), 3.78 − 3.72 (m, 4H), 3.44 − 3.34 (m, 7H), 1.84 (s, 3H). ^13^C NMR (150 MHz, DMSO-d_6_) δ 170.5, 162.4, 154.8, 149.1 (d, JC-F = 256.9 Hz), 145.9, 145.8, 133.1, 133.0, 130.0, 115.6, 115.4, 72.6, 63.1, 52.5, 47.7, 43.7, 41.9, 30.6, 22.9, 7.7. HRMS (ESI) (positive mode) *m*/*z* calculated for C_22_H_25_FN_8_O_3_: 468.49; Found: 469.201.

*(S)-N-((3-(5-fluoro-6-(4-(2-(4-methylpiperidin-1-yl)pyrimidin-4-yl)piperazin-1-yl)pyridin-3-yl)-2-oxooxazolidin-5-yl)methyl)acetamide* (**6f**).

Compound **6f** was a pink solid; yield 64.4%. m. p. 198.1–200.5 °C.^1^H NMR (600 MHz, DMSO-d_6_) δ 8.24 (t, J = 6.0 Hz, 1H), 8.13 (d, J = 2.4 Hz, 1H), 7.93 (dd, J = 14.4, 2.4 Hz, 1H), 7.89 (d, J = 6.0 Hz, 1H), 6.08 (d, J = 6.0 Hz, 1H), 4.79 − 4.72 (m, 1H), 4.62 − 4.59 (m, 2H), 4.13 − 4.10 (m, 1H), 3.74 − 3.72 (m, 1H), 3.68 − 3.66 (m, 4H), 3.43 − 3.41 (m, 2H), 3.41 − 3.37 (m, 4H), 2.77 − 2.73 (m, 2H), 1.84 (s, 3H), 1.64 − 1.56 (m, 2H), 1.04 − 0.98 (m, 2H), 0.91 (d, J = 6.6 Hz, 3H). ^13^C NMR (150 MHz, DMSO-d_6_) δ 170.5, 162.7, 161.4, 157.1, 154.8, 149.1 (d, JC-F = 256.3 Hz), 146.0, 133.0, 129.9, 115.5, 115.4, 93.2, 72.6, 47.7, 44.0, 43.6, 41.9, 34.1, 31.3, 22.9, 22.4. HRMS (ESI) (positive mode) *m*/*z* calculated for C_25_H_33_FN_8_O_3_: 512.59; Found: 513.208.

*(S)-N-((3-(5-fluoro-6-(4-(2-morpholinopyrimidin-4-yl)piperazin-1-yl)pyridin-3-yl)-2-oxooxazolidin-5-yl)methyl)acetamide* (**6g**)

Compound **6g** was a white solid; yield 46.1%. m. p. 191.3–192.1 °C. ^1^H NMR (600 MHz, DMSO-d_6_) δ 8.24 (t, J = 6.0 Hz, 1H), 8.13 (d, J = 2.4 Hz, 1H), 7.96 − 7.90 (m, 2H), 6.17 (d, J = 6.0 Hz, 1H), 4.79 − 4.72 (m, 1H), 4.13 − 4.10 (m, 1H), 3.74 − 3.72 (m, 1H), 3.70 − 3.69 (m, 4H), 3.64 − 3.63 (m, 8H), 3.47 − 3.35 (m, 6H), 1.84 (s, 3H).^13^C NMR (150 MHz, DMSO-d_6_) δ 170.5, 162.6, 161.6, 157.1, 154.8, 150.0 (d, JC-F = 256.7 Hz), 146.0, 133.0, 129.9, 115.5, 115.4, 94.1, 72.6, 66.6, 47.7, 44.4, 43.6, 41.9, 22.9. HRMS (ESI) (positive mode) m/z calculated for C_23_H_29_FN_8_O_4_: 500.54; Found: 501.176.

*(S)-N-((3-(5-fluoro-6-(4-(2-((3-morpholinopropyl)amino)pyrimidin-4-yl)piperazin-1-yl)pyridin-3-yl)-2-oxooxazolidin-5-yl)methyl)acetamide* (**6h**)

Compound **6h** was brown oil; yield 48.7%. ^1^H NMR (600 MHz, DMSO-d_6_) δ 8.28 (t, J = 6.0 Hz, 1H), 8.12 (d, J = 2.4 Hz, 1H), 7.92 (dd, J = 14.4, 2.4 Hz, 1H), 7.81 (d, J = 6.0 Hz, 1H), 6.57 (s, 1H), 6.03 (d, J = 6.0 Hz, 1H), 4.77 − 4.73 (m, 1H), 4.12 − 4.09 (m, 1H), 3.75 − 3.73 (m, 1H), 3.67 − 3.65 (m, 4H), 3.57 − 3.56 (m, 4H), 3.44 − 3.42 (m, 4H), 3.38 − 3.35 (m, 4H), 3.27 − 3.23 (m, 2H), 2.33 − 2.30 (m, 4H), 1.85 (s, 3H), 1.66 − 1.64 (m, 2H). ^13^C NMR (150 MHz, DMSO-d_6_) δ 170.6, 162.7, 162.3, 157.2, 154.8, 149.1 (d, JC-F = 257.1 Hz), 145.9, 132.9, 129.8, 115.5, 115.3, 72.6, 66.7, 56.7, 53.8, 47.7, 46.1, 43.5, 41.9, 26.5, 22.9, 7.6. HRMS (ESI) (positive mode) *m*/*z* calculated for C_26_H_35_FN_8_O_4_: 557.63; Found: 558.256.

*(S)-N-((3-(5-fluoro-6-(4-(2-(phenylamino)pyrimidin-4-yl)piperazin-1-yl)pyridin-3-yl)-2-oxooxazolidin-5-yl)methyl)acetamide* (**6i**)

Compound **6i** was a white solid; yield 43%. m. p. 190.6–194.4 °C.^1^H NMR (600 MHz, DMSO-d_6_) δ 9.17 (s, 1H), 8.24 (t, J = 6.0 Hz, 1H), 8.14 (d, J = 2.4 Hz, 1H), 8.00 (d, J = 6.0 Hz, 1H), 7.95 (dd, J = 14.4, 2.4 Hz, 1H), 7.73 − 7.68 (m, 2H), 7.30 − 7.24 (m, 2H), 6.92 (t, J = 7.2 Hz, 1H), 6.36 (d, J = 6.0 Hz, 1H), 4.79 − 4.72 (m, 1H), 4.14 − 4.11 (m, 1H), 3.79 − 3.77 (m, 4H), 3.75 − 3.72 (m, 1H), 3.48 − 3.39 (m, 6H), 1.84 (s, 3H). ^13^C NMR (150 MHz, DMSO-d_6_) δ 170.5, 165.2, 162.5, 154.8, 149.1 (d, JC-F = 256.2 Hz), 145.7, 141.2, 140.8, 133.0, 129.8, 128.9, 121.8, 119.4, 117.3, 115.6, 95.8, 72.6, 47.7, 43.8, 41.9, 22.9. HRMS (ESI) (positive mode) *m*/*z* calculated for C_25_H_27_FN_8_O_3_: 506.54; Found: 507.161.

*(S)-N-((3-(6-(4-(2-(benzylamino)pyrimidin-4-yl)piperazin-1-yl)-5-fluoropyridin-3-yl)-2-oxooxazolidin-5-yl)methyl)acetamide* (**6j**)

Compound **6j** was a white solid; yield 41.5%. m. p. 202.7–204.6 °C.^1^H NMR (600 MHz, DMSO-d_6_) δ 8.24 (t, J = 6.0 Hz, 1H), 8.13 (d, J = 2.4 Hz, 1H), 7.93 (dd, J = 14.4, 2.4 Hz, 1H), 7.82 (d, J = 6.0 Hz, 1H), 7.32 − 7.24 (m, 5H), 7.22 − 7.16 (m, 1H), 6.07 (d, J = 6.0 Hz, 1H), 4.79 − 4.72 (m, 1H), 4.14 − 4.10 (m, 1H), 3.74 − 3.72 (m, 1H), 3.74 − 3.64 (m, 4H), 3.47 − 3.38 (m, 2H), 3.36 − 3.34 (m, 4H), 1.84 (s, 3H). ^13^C NMR (150 MHz, DMSO-d_6_) δ 170.5, 162.6, 162.3, 154.8, 150.0 (d, JC-F = 256.7 Hz), 146.0, 145.9, 141.7, 133.0, 129.8, 128.5, 127.6, 126.8, 115.5, 115.4, 72.6, 47.8, 44.5, 43.5, 41.9, 22.9. HRMS (ESI) (positive mode) *m*/*z* calculated for C_26_H_29_FN_8_O_3_: 520.57; Found: 521.185.

*(S)-N-((3-(5-fluoro-6-(4-(2-(naphthalen-1-ylamino)pyrimidin-4-yl)piperazin-1-yl)pyridin-3-yl)-2-oxooxazolidin-5-yl)methyl)acetamide* (**6k**)

Compound **6k** was a brown solid; yield 59.3%. m. p. 221.8–223.1 °C.1H NMR (600 MHz, DMSO-d6) δ 8.95 (s, 1H), 8.24 (t, J = 6.0 Hz, 1H), 8.14–8.07 (m, 2H), 7.95 − 7.88 (m, 3H), 7.79 (d, J = 7.2 Hz, 1H), 7.67 (d, J = 8.2 Hz, 1H), 7.51–7.44 (m, 3H), 6.29 (d, J = 6.0 Hz, 1H), 4.77–4.72 (m, 1H), 4.15–4.06 (m, 1H), 3.76–3.70 (m, 1H), 3.69–3.64 (m, 4H), 3.43–3.40 (m, 2H), 3.39–3.36(m, 4H), 1.84 (s, 3H). ^13^C NMR (150 MHz, DMSO-d_6_) δ 170.6, 162.1, 154.8, 152.4, 149.1 (d, JC-F = 257.1 Hz), 146.0, 145.7, 134.9, 134.4, 133.0, 130.0, 128.7, 128.6, 126.4, 126.1, 126.0, 125.1, 123.4, 121.7, 115.6, 115.4, 95.8, 72.6, 63.1, 52.5, 22.9, 7.7. HRMS (ESI) (positive mode) *m*/*z* calculated for C_29_H_29_FN_8_O_3_: 556.60; Found: 557.197.

*(S)-N-((3-(5-fluoro-6-(4-(2-(quinolin-5-ylamino)pyrimidin-4-yl)piperazin-1-yl)pyridin-3-yl)-2-oxooxazolidin-5-yl)methyl)acetamide* (**6l**)

Compound **6l** was a white solid; yield 43%. m. p. 226.2–232.2 °C.^1^H NMR (600 MHz, DMSO-d_6_) δ 9.15 (s, 1H), 8.87 (dd, J = 4.2, 1.8 Hz, 1H), 8.53 − 8.48 (m, 1H), 8.24 (t, J = 6.0 Hz, 1H), 8.15 − 8.12 (m, 1H), 7.99 − 7.90 (m, 2H), 7.87 (dd, J = 7.2, 1.2 Hz, 1H), 7.77 − 7.71 (m, 2H), 7.489 − 7.47 (m, 1H), 6.33 (d, J = 6.0 Hz, 1H), 4.79 − 4.72 (m, 1H), 4.13 − 4.10 (m, 1H), 3.74 − 3.72 (m, 1H), 3.68 − 3.66 (m, 4H), 3.43 − 3.41 (m, 2H), 3.39 − 3.37 (m, 4H), 1.84 (s, 3H). ^13^C NMR (150 MHz, DMSO-d_6_) δ 170.5, 162.5, 161.1, 157.3, 154.8, 149.7 (d, JC-F = 259.1 Hz), 148.3, 145.9, 136.9, 133.0, 132.6, 129.9, 129.6, 124.7, 123.6, 120.7, 120.6, 115.6, 115.4, 107.8, 95.9, 72.6, 47.7, 43.5, 41.9, 22.9. HRMS (ESI) (positive mode) *m*/*z* calculated for C_28_H_28_FN_9_O_3_: 557.59; Found: 558.207.

*(S)-N-((3-(6-(4-(2-((6-chloropyridin-3-yl)amino)pyrimidin-4-yl)piperazin-1-yl)-5-fluoropyridin-3-yl)-2-oxooxazolidin-5-yl)methyl)acetamide* (**6m**)

Compound **6m** was a white solid; yield 45.8%. m. p. 134.4–135.6 °C.^1^H NMR (600 MHz, DMSO-d_6_) δ 9.46 (s, 1H), 8.74 (d, J = 3.0 Hz, 1H), 8.27 − 8.19 (m, 2H), 8.14 (d, J = 2.4 Hz, 1H), 8.05 (d, J = 6.0 Hz, 1H), 7.94 (dd, J = 14.4, 2.4 Hz, 1H), 7.40 (d, J = 9.0 Hz, 1H), 6.40 (d, J = 6.0 Hz, 1H), 4.79 − 4.72 (m, 1H), 4.14 − 4.11 (m, 1H), 3.78 − 3.71 (m, 5H), 3.47 − 3.42 (m, 6H), 1.84 (s, 3H). ^13^C NMR (150 MHz, DMSO-d_6_) δ 170.5, 162.5, 159.5, 157.1, 154.8, 149.1 (d, JC-F = 256.5 Hz), 145.8, 141.3, 140.2, 137.9, 133.0, 129.9, 129.2, 124.2, 115.6, 115.4, 96.6, 72.6, 47.6, 43.7, 41.9, 22.9. HRMS (ESI) (positive mode) *m*/*z* calculated for C_24_H_25_ClFN_9_O_3_: 541.97; Found: 542.132.

Raw data for the above products are presented in Appendix A.

#### 3.2.4. General Procedure for the Synthesis of **7a** and **7c**-**n**

A solution of compound **4** (130 mg, 0.3 mmol) in DCM (5 mL) at 0 °C was dropwise added to TFA (1 mL) and then stirred for 2 h. After the reaction was complete, TEA was added to the solution at 0 °C to adjust pH. The filtrate was concentrated in vacuo. To a solution of the concentrate in ethanol (3 mL) was added TEA (83 μL, 1 mmol) and pyrimidine derivative (0.4 mmol), and then stirred at reflux overnight. After the reaction was complete and concentrated, the mixture was extracted with DCM (5 mL × 3). The organic phase was washed with brine and concentrated in vacuo. The residue was purified by silica gel column chromatography (DCM/MeOH = 30:1) to yield compounds **7a** and **7c-n**.

*(S)-N-((3-(6-(4-(6-chloropyrimidin-4-yl)piperazin-1-yl)-5-fluoropyridin-3-yl)-2-oxooxazolidin-5-yl)methyl)acetamide* (**7a**)

Compound **7a** was a white solid; yield 45.1%. m. p. 190.0–191.1 °C.^1^H NMR (600 MHz, DMSO-d_6_) δ 8.37 (s, 1H), 8.24 (t, J = 6.0 Hz, 1H), 8.14 (d, J = 2.4 Hz, 1H), 7.94 (dd, J = 14.4, 2.4 Hz, 1H), 7.02 (s, 1H), 4.79 − 4.72 (m, 1H), 4.14 − 4.10 (m, 1H), 3.81 − 3.75 (m, 4H), 3.74 − 3.71 (m, 1H), 3.43 − 3.42 (m, 6H), 1.84 (s, 3H). ^13^C NMR (150 MHz, DMSO-d_6_) δ 170.5, 162.7, 159.7, 158.5, 154.8, 148.2 (d, JC-F = 256.7 Hz), 145.6, 133.0, 129.99, 115.6, 102.3, 72.6, 47.6, 47.5, 43.8, 41.9, 22.9. HRMS (ESI) (positive mode) *m*/*z* calculated for C_19_H_21_ClFN_7_O_3_: 449.87; Found: 450.106.

*(S)-N-((3-(5-fluoro-6-(4-(5-methylpyrimidin-2-yl)piperazin-1-yl)pyridin-3-yl)-2-oxooxazolidin-5-yl)methyl)acetamide* (**7c**)

Compound **7c** was a white solid; yield 49.4%. m. p. 195.1–197.3 °C.^1^H NMR (600 MHz, DMSO-d_6_) δ 8.26 − 8.23 (m, 3H), 8.13 (d, J = 2.4 Hz, 1H), 7.93 (dd, J = 14.4, 2.4 Hz, 1H), 4.77 − 4.73 (m, 1H), 4.13 − 4.10 (m, 1H), 3.85 − 3.80 (m, 4H), 3.76 − 3.70 (m, 1H), 3.43 − 3.37 (m, 6H), 2.10 (s, 3H), 1.84 (s, 3H). ^13^C NMR (150 MHz, DMSO-d_6_) δ 160.7, 158.2, 154.8, 149.2 (d, JC-F = 255.3 Hz), 146.1, 133.0, 130.0, 119.1, 115.5, 115.4, 72.6, 47.9, 46.2, 43.9, 22.9, 14.1, 9.1. HRMS (ESI) (positive mode) *m*/*z* calculated for C_20_H_24_FN_7_O_3_: 429.46; Found: 430.181.

*(S)-N-((3-(6-(4-(5-bromopyrimidin-2-yl)piperazin-1-yl)-5-fluoropyridin-3-yl)-2-oxooxazolidin-5-yl)methyl)acetamide* (**7d**)

Compound **7d** was a white solid; yield 47.6%. m. p. 196.1–196.3 °C.^1^H NMR (600 MHz, DMSO-d_6_) δ 8.50 (s, 2H), 8.24 (t, J = 6.0 Hz, 1H), 8.13 (d, J = 2.4 Hz, 1H), 7.94 (dd, J = 14.4, 2.4 Hz, 1H), 4.77 − 4.73 (m, 1H), 4.14 − 4.10 (m, 1H), 3.88 − 3.83 (m, 4H), 3.74 − 3.71 (m, 1H), 3.43 − 3.33 (m, 6H), 1.84 (s, 3H). ^13^C NMR (150 MHz, DMSO-d_6_) δ 170.5, 166.3, 160.0, 158.5, 154.8, 150.8 (d, JC-F = 253.7 Hz), 133.0, 129.9, 115.3, 106.1, 72.6, 47.6, 43.8, 41.9, 39.0, 22.9. HRMS (ESI) (positive mode) *m*/*z* calculated for C_19_H_21_BrFN_7_O_3_: 494.33; Found: 496.095.

*(S)-N-((3-(6-(4-(2-aminopyrimidin-4-yl)piperazin-1-yl)-5-fluoropyridin-3-yl)-2-oxooxazolidin-5-yl)methyl)acetamide* (**7e**)

Compound **7e** was a white solid; yield 60.9%. m. p. 195.6–196.7 °C.^1^H NMR (600 MHz, DMSO-d_6_) δ 8.28 − 8.25 (m, 1H), 8.15 (d, J = 2.4 Hz, 1H), 7.97 − 7.88 (m, 3H), 7.89 (dd, J = 7.8, 2.4 Hz, 1H), 6.57 (d, J = 7.8 Hz, 1H), 4.78 − 4.74 (m, 1H), 4.14 − 4.11 (m, 1H), 3.75 − 3.72 (m, 1H), 3.47 − 3.46 (m, 4H), 3.43 − 3.41 (m, 4H), 1.84 (s, 3H). ^13^C NMR (150 MHz, DMSO-d_6_) δ 170.5, 161.7, 155.1, 154.8, 149.1 (d, JC-F = 256.4 Hz), 145.4, 143.4, 133.0, 130.1, 115.6, 115.5, 95.4, 72.7, 47.6, 47.5, 41.9, 22.9. HRMS (ESI) (positive mode) *m*/*z* calculated for C_19_H_23_FN_8_O_3_: 430.44; Found: 431.144.

*(S)-N-((3-(6-(4-(4-aminopyrimidin-2-yl)piperazin-1-yl)-5-fluoropyridin-3-yl)-2-oxooxazolidin-5-yl)methyl)acetamide* (**7f**)

Compound **7f** was a white solid; yield 60.9%. m. p. 196.6–197.7 °C.^1^H NMR (600 MHz, DMSO-d_6_) δ 8.25 (t, J = 6.0 Hz, 1H), 8.14 (d, J = 2.4 Hz, 1H), 7.94 (dd, J = 14.4, 2.4 Hz, 1H), 7.77 (d, J = 6.0 Hz, 1H), 7.48 − 7.10 (m, 2H), 5.94 (d, J = 6.0 Hz, 1H), 4.79 − 4.72 (m, 1H), 4.13 − 4.10 (m, 1H), 3.80 − 3.78 (m, 4H), 3.75 − 3.72 (m, 1H), 3.43 − 3.40 (m, 6H), 1.84 (s, 3H). ^13^C NMR (150 MHz, DMSO-d_6_) δ 170.5, 164.5, 154.8, 149.1 (d, JC-F = 256.7 Hz), 145.9, 138.1, 133.0, 130.0, 128.5, 126.0, 115.6, 72.6, 47.7, 43.9, 41.9, 22.9, 21.3. HRMS (ESI) (positive mode) *m*/*z* calculated for C_19_H_23_FN_8_O_3_: 430.44; Found: 431.141. 

*(S)-N-((3-(6-(4-(4,6-dichloropyrimidin-2-yl)piperazin-1-yl)-5-fluoropyridin-3-yl)-2-oxooxazolidin-5-yl)methyl)acetamide*(**7g**)and *(S)-N-((3-(6-(4-(2,6-dichloropyrimidin-4-yl)piperazin-1-yl)-5-fluoropyridin-3-yl)-2-oxooxazolidin-5-yl)methyl)acetamide* (**7h**)

Compounds **7g** and **7h** were both white solids, yields were 13.8% and 44.7% respectively, m. p. 191.3–192.5 °C and m. p. 195.2–197.1 °C respectively.

Compound **7g**, ^1^H NMR (600 MHz, DMSO-d_6_) δ 8.24 (t, J = 6.0 Hz, 1H), 8.14 (d, J = 2.4 Hz, 1H), z7.94 (dd, J = 14.4, 2.4 Hz, 1H), 6.98 (s, 1H), 4.77 − 4.73 (m, 1H), 4.14 − 4.10 (m, 1H), 3.88 − 3.84 (m, 4H), 3.75 − 3.72 (m, 1H), 3.44 − 3.41 (m, 6H), 1.84 (s, 3H). ^13^C NMR (150 MHz, DMSO-d_6_) δ 170.5, 161.6, 160.5, 154.8, 148.2 (d, J = 256.7 Hz), 145.7, 132.9, 130.0, 115.4, 108.2, 72.6, 47.6, 47.5, 43.9, 41.9, 22.9. HRMS (ESI) (positive mode) *m*/*z* calculated for C_19_H_20_Cl_2_FN_7_O_3_: 483.10; Found: 484.103.

Compound **7h**, ^1^H NMR (600 MHz, DMSO-d_6_) δ 8.24 (t, J = 6.0 Hz, 1H), 8.14 (d, J = 2.4 Hz, 1H), 7.94 (dd, J = 14.4, 2.4 Hz, 1H), 7.08 (s, 1H), 4.77 − 4.74 (m, 1H), 4.13 − 4.10 (m, 1H), 3.87 − 3.71 (m, 5H), 3.44 − 3.42 (m, 6H), 1.84 (s, 3H). ^13^C NMR (150 MHz, DMSO-d_6_) δ 170.5, 163.4, 159.6, 159.0, 154.8, 149.1 (d, JC-F = 257.1 Hz), 145.5, 133.0, 130.0, 115.4, 101.4, 72.6, 47.6, 47.4, 43.7, 41.9, 22.9. HRMS (ESI) (positive mode) *m*/*z* calculated for C_19_H_20_Cl_2_FN_7_O_3_: 483.10; Found: 484.104.

*(S)-N-((3-(6-(4-(2-chloro-5-fluoropyrimidin-4-yl)piperazin-1-yl)-5-fluoropyridin-3-yl)-2-oxooxazolidin-5-yl)methyl)acetamide* (**7i**)

Compound **7i** was a white solid; yield 70.1%. m. p. 204.4–205.5 °C.^1^H NMR (600 MHz, DMSO-d_6_) δ 8.25 − 8.21 (m, 2H), 8.14 (d, J = 2.4 Hz, 1H), 7.94 (dd, J = 14.4, 2.4 Hz, 1H), 4.79 − 4.72 (m, 1H), 4.14 − 4.10 (m, 1H), 3.90 − 3.86 (m, 4H), 3.75 − 3.71 (m, 1H), 3.48 − 3.46 (m, 4H), 3.43 − 3.41 (m, 2H), 1.84 (s, 3H). ^13^C NMR (150 MHz, DMSO-d_6_) δ 170.5, 154.8, 153.5, 152.5, 149.1 (d, JC-F = 257.0 Hz), 147.1 (d, JC-F = 256.4 Hz), 145.6, 144.7, 133.0, 130.0, 115.6, 72.6, 47.6, 45.8, 41.9, 40.5, 22.9. HRMS (ESI) (positive mode) *m*/*z* calculated for C_19_H_20_ClF_2_N_7_O_3_: 467.86; Found: 468.113.

*(S)-N-((3-(6-(4-(2,5-dichloropyrimidin-4-yl)piperazin-1-yl)-5-fluoropyridin-3-yl)-2-oxooxazolidin-5-yl)methyl)acetamide* (**7j**)

Compound **7j** was a white solid; yield 48.6%. m. p. 193.5–194.2 °C.^1^H NMR (600 MHz, DMSO-d_6_) δ 8.35 (s, 1H), 8.24 (t, J = 6.0 Hz, 1H), 8.14 (d, J = 2.4 Hz, 1H), 7.94 (dd, J = 14.4, 2.4 Hz, 1H), 4.77 − 4.73 (m, 1H), 4.14 − 4.10 (m, 1H), 3.91 − 3.86 (m, 4H), 3.75 − 3.72 (m, 1H), 3.51 − 3.46 (m, 4H), 3.43 − 3.41 (m, 2H), 1.84 (s, 3H). ^13^C NMR (150 MHz, DMSO-d_6_) δ 170.5, 160.4, 159.1, 157.1, 154.8, 148.3 (d, JC-F = 256.7 Hz), 133.0, 130.0, 115.5, 115.1, 72.6, 47.6, 47.6, 47.0, 41.9, 22.9. HRMS (ESI) (positive mode) *m*/*z* calculated for C_19_H_20_Cl_2_FN_7_O_3_: 483.10; Found: 484.099. 

*(S)-N-((3-(6-(4-(5-bromo-2-chloropyrimidin-4-yl)piperazin-1-yl)-5-fluoropyridin-3-yl)-2-oxooxazolidin-5-yl)methyl)acetamide* (**7k**)

Compound **7k** was a white solid; yield 82.8%. m. p. 196.7–197.6 °C. ^1^H NMR (600 MHz, DMSO-d_6_) δ 8.46 (s, 1H), 8.24 (t, J = 6.0 Hz, 1H), 8.14 (d, J = 2.4 Hz, 1H), 7.94 (dd, J = 14.4, 2.4 Hz, 1H), 4.77 − 4.73 (m, 1H), 4.14 − 4.10 (m, 1H), 3.87 − 3.83 (m, 4H), 3.75 − 3.72 (m, 1H), 3.50 − 3.46 (m, 4H), 3.43 − 3.41 (m, 2H), 1.84 (s, 3H). ^13^C NMR (150 MHz, DMSO-d_6_) δ 170.5, 162.7, 161.8, 157.8, 154.8, 149.1 (d, JC-F = 257.1 Hz), 145.6, 133.0, 130.0, 115.5, 104.1, 72.6, 47.6, 47.6, 47.3, 41.9, 22.9. HRMS (ESI) (positive mode) *m*/*z* calculated for C_19_H_20_BrClFN_7_O_3_: 528.77; Found: 530.054.

*(S)-N-((3-(6-(4-(2-chloro-5-methylpyrimidin-4-yl)piperazin-1-yl)-5-fluoropyridin-3-yl)-2-oxooxazolidin-5-yl)methyl)acetamide* (**7l**)

Compound **7l** was a white solid; yield 58.9%. m. p. 198.8–200.8 °C. ^1^H NMR (600 MHz, DMSO-d_6_) δ 8.24 (t, J = 6.0 Hz, 1H), 8.14 (d, J = 2.4 Hz, 1H), 8.06 (s, 1H), 7.93 (dd, J = 14.4, 2.4 Hz, 1H), 4.79 − 4.72 (m, 1H), 4.14 − 4.10 (m, 1H), 3.74 − 3.72 (m, 1H), 3.70 − 3.65 (m, 4H), 3.48 − 3.44 (m, 4H), 3.43 − 3.41 (m, 2H), 2.24 (s, 3H), 1.84 (s, 3H). ^13^C NMR (150 MHz, DMSO-d_6_) δ 170.5, 165.0, 160.0, 157.2, 154.8, 149.1 (d, JC-F = 257.1 Hz), 145.8, 133.0, 130.0, 116.2, 115.4, 72.6, 47.8, 47.6, 46.9, 41.9, 22.9, 17.3. HRMS (ESI) (positive mode) *m*/*z* calculated for C_20_H_23_ClFN_7_O_3_: 463.90; Found: 464.117.

*Ethyl(S)-4-(4-(5-(5-(acetamidomethyl)-2-oxooxazolidin-3-yl)-3-fluoropyridin-2-yl)piperazin-1-yl)-2-(methylthio)pyrimidine-5-carboxylate* (**7m**)

Compound **7m** was a white solid; yield 50.6%. m. p. 210.1–213.1 °C. ^1^H NMR (600 MHz, DMSO-d_6_) δ 8.45 (s, 1H), 8.26 (t, J = 6.0 Hz, 1H), 8.11 (d, J = 2.4 Hz, 1H), 7.92 (dd, J = 14.4, 2.4 Hz, 1H), 4.79 − 4.72 (m, 1H), 4.36 − 4.17 (m, 2H), 4.14 − 4.09 (m, 1H), 3.76 − 3.72 (m, 1H), 3.68 − 3.66 (m, 4H), 3.46 − 3.44 (m, 4H), 3.43 − 3.41 (m, 2H), 2.48 (s, 3H), 1.85 (s, 3H), 1.37 − 1.25 (m, 3H). ^13^C NMR (150 MHz, DMSO-d_6_) δ 172.6, 170.5, 165.9, 159.8, 159.3, 154.8, 149.0 (d, JC-F = 257.0 Hz), 145.6, 132.9, 129.8, 115.3, 105.7, 72.6, 61.4, 47.6, 47.5, 47.3, 41.9, 22.9, 14.5, 14.1. HRMS (ESI) (positive mode) *m*/*z* calculated for C_23_H_28_FN_7_O_5_S: 533.58; Found: 534.132.

*(S)-N-((3-(5-fluoro-6-(4-(2,5,6-trichloropyrimidin-4-yl)piperazin-1-yl)pyridin-3-yl)-2-oxooxazolidin-5-yl)methyl)acetamide* (**7n**)

Compound **7n** was a white solid; yield 71.4%. m. p. 210.0–211.1 °C. ^1^H NMR (600 MHz, DMSO-d_6_) δ 8.24 (t, J = 6.0 Hz, 1H), 8.14 (d, J = 2.4 Hz, 1H), 7.94 (dd, J = 14.4, 2.4 Hz, 1H), 4.79 − 4.72 (m, 1H), 4.13 − 4.10 (m, 1H), 3.92 − 3.81 (m, 4H), 3.75 − 3.72 (m, 1H), 3.52 − 3.47 (m, 4H), 3.44 − 3.39 (m, 2H), 1.84 (s, 3H). ^13^C NMR (150 MHz, DMSO-d_6_) δ 170.5, 161.9, 159.0, 155.1, 154.8, 149.1 (d, JC-F = 255.8 Hz), 145.6, 133.0, 129.8, 115.4, 112.4, 72.6, 47.8, 47.6, 47.5, 41.9, 22.9. HRMS (ESI) (positive mode) *m*/*z* calculated for C_19_H_19_Cl_3_FN_7_O_3_: 518.76; Found: 520.066.

Raw data for the above products are presented in Appendix A.

### 3.3. MIC Determination

The antibacterial activity of the synthesized derivatives was determined by the broth dilution method [42]. The strains, including *Staphylococcus aureus* (ATCC25923), *Streptococcus pneumoniae* (ATCC49619), *Enterococcus faecalis* (ATCC29212), *Bacillus subtilis* (ATCC6633), *Escherichia coli* (ATCC25922), *Listeria monocytogenes* (ATCC19111), *Staphylococcus xylosus (ATCC35924)*, methicillin-resistant *Staphylococcus aureus*, vancomycin-resistant *Enterococcus faecalis*, linezolid-resistant *Staphylococcus aureus,* and linezolid-resistant *Streptococcus pneumoniae*, were incubated in Mueller-Hinton (MH) medium at 37 °C to mid-logarithm (OD_600_ = 0.5). The bacteria were diluted to 10^5^ CFU/ mL and added to the 96-well plate, followed by a series of diluted synthesized derivatives (from 128 to 0.25 µg/mL). After incubation at 37 °C for 16–18 h, the minimum inhibitory concentration (MIC) value was the minimum drug concentration without bacterial growth. 

### 3.4. Molecular Docking Studies

The 3D structure of the 50S ribosomal subunit (PDB code: 3CPW) [43] was obtained from the Protein Data Bank and processed by PyMOL 2.5. The original ligand and protein were deleted, and only the required RNA chains were retained and imported into the Auto-Dock for use. The ligand was drawn with ChemOffice2010 Version and imported into the Auto-Dock for later use. The Auto-Dock 4.2.6^®^ software was used for the molecular docking process, and the obtained results were imported into PyMOL 2.5 software in the form of complexes for visual analysis.

### 3.5. Inhibition of Biofilm Formation Assay

Anti-biofilm activity inhibits biofilm formation and was measured by the crystal violet method [44]. The strains to be tested were placed in a test tube containing 5 mL Tryptic Soy Broth (TSB) and incubated at 37 °C for 24 h. Then the suspension was diluted to 10^6^ CFU/mL and added to a sterile 96-well culture plate, filled with 100 µL per well. All compounds were added to the well according to the selected concentration gradient and incubated at 37 °C for 24 h. After the biofilm was grown, the culture medium was removed from each well, washed twice with sterile PBS, fixed with methanol, and stained with 150 µL 0.1% crystal violet solution at room temperature. Remove the excess solution, wash it twice with water, and add 125 µL 33% acetic acid to each dyeing well for 5 min to dissolve the dye. The microplate reader was used to read at 600 nm to assess the minimum concentration of biofilm inhibition.

### 3.6. Cytotoxicity Assay

The MTT method was used to detect the effect of typical derivatives on Hela cell viability [45]. Cells (5 × 10^4^ cells/well) were added to a 96-well plate for 24 h in humidified 5% (V/V) CO_2_/ air at 37 °C. A series of liquid medicines (8, 16, 32, 64, 128, 256, 500, and 1000 µg/mL) were added and incubated for 48 h. Cells treated with equal volumes of DMSO were used as controls. Add 10 µL MTT solution (0.5%) to each well and incubate for 4 h at 37 °C under dark conditions. The culture medium in all wells was discarded. Then 100 µL DMSO was quickly added to each wall and shaken at low speed to dissolve the formed crystals. The absorbance was measured at 570 nm. The cell viability was calculated as follows: cell viability (%) = (treatment sample OD_570_ − empty OD_570_)/(control OD_570_ − empty OD_570_).

## 4. Conclusions

In summary, a library of 28 novel 3-(5-fluoropyridine-3-yl)-2-oxazolidinone derivatives was designed, synthesized, and evaluated for their antibacterial properties. The results show that most of the synthesized compounds have potential antibacterial activity against gram-positive bacteria. Amongst them, compounds **7i-l** exhibited a better antibacterial effect. The molecular docking results of compounds **7j** and PTC were studied to predict the mechanism of action. Further results demonstrated that these compounds have excellent ability to inhibit biofilm formation and meager cytotoxicity. These results provide a basis and reference for the discovery of novel antibacterial compounds and the development of new drugs.

## Data Availability

Not applicable.

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
