# Peer review of "Optimization and Antibacterial Evaluation of Novel 3-(5-Fluoropyridine-3-yl)-2-oxazolidinone Derivatives Containing a Pyrimidine Substituted Piperazine"

_molecules, 2023, doi:10.3390/molecules28114267_

Round 1
Reviewer 1 Report
The manuscript “Optimization and Antibacterial Evaluation of Novel 3-(5-Fluoropyridine-3-yl)-2-Oxazolidinone Derivatives Containing a Pyrimidine Substituted Piperazine” reports a comprehensive research on the novel heterocyclic ensembles and their biological properties. The manuscript can be accepted by Molecules after major revision:
1. Figure 1: compounds 1 and 2 should be labeled.
2. “The allyl acetone structure in these” this is not allyl, this is vinyl.
3. “the unique electronegativity of pyrimidines and the ability to form hydrogen bonds” Where the uniqueness does come from? This adjective should be omitted or a reference is required.
4. Figure 1: where is X on the scheme?
5. Scheme 1: arrow from 3 ro 4 should be different. When you do not disclose the exact stages it should be two or three arrows diagonally above each other.
6. R1, R2, X1, X2, and etc. – the digit must be superscript.
7. Why the 50S ribosomal subunit from Haloarcula Marismortui was chosen for molecular docking?
8. 1H NMRs of compounds 5, 6a, 6c, 6d, 6e, 6h, 6j, 6k, 7a, 7b, 7c, 7f, 7l, 7m, 7n show signal of impurities (solvents and side-products). These could affect the results of bioevaluations. The data on purity and potent impurities in the tested compounds must be added to the experimental section.
Author Response
- Figure 1: compounds 1 and 2 should be labeled.
Reply: The action strains of compounds 1 and 2 have been labeled. And the references of these compounds have been given.
- “The allyl acetone structure in these” this is not allyl, this is vinyl.
Reply:We have changed the ally acetone to vinyl.
- “the unique electronegativity of pyrimidines and the ability to form hydrogen bonds” Where the uniqueness does come from? This adjective should be omitted or a reference is required.
Reply:The literatures about “The unique electronegativity of pyrimidines and the ability to form hydrogen bonds” have been cited, such as 20-22.
- Figure 1: where is X on the scheme?
Reply:The X on the scheme2 has been superscripted.
- Scheme 1: arrow from 3 ro 4 should be different. When you do not disclose the exact stage it should be two or three arrows diagonally above each other.
Reply:The arow from 3 or 4 in scheme 1has been modified.
- R1, R2, X1, X2, and etc. – the digit must be superscript.
Reply:The digit in the R1、R2、X1 have been superscripted.
- Why the 50S ribosomal subunit from Haloarcula Marismortui was chosen for molecular docking?
Reply: In many literatures about the synthesis of linezolid derivatives, the above ribosomes are used for molecular docking, such as 38 and 39.
- 1H NMRs of compounds 5, 6a, 6c, 6d, 6e, 6h, 6j, 6k, 7a, 7b, 7c, 7f, 7l, 7m, 7n show signal of impurities (solvents and side-products). These could affect the results of bioevaluations. The data on purity and potent impurities in the tested compounds must be added to the experimental section.
Reply:There is a little organic solvent residue in the compounds, we have marked the corresponding solvents peak in the spectrum, and we can determine that the compounds used in the experiment are pure by high-performance liquid chromatography.

Reviewer 2 Report
The paper ‘Optimization and Antibacterial Evaluation of Novel
3-(5-Fluoropyridine-3-yl)-2-Oxazolidinone Derivatives Containing a Pyrimidine Substituted Piperazine’ describes the preparation and early biological assessment as antibiotics of a set of oxazolidinone derivatives. The compounds are close analogues to linezolid and are suggested to have the same mode of action as predicted by computational chemistry docking to the ribosome target.
The synthetic work is straightforward and clearly described and of good quality. The biological assessment is fine, however, limited to the very first biological questions you may ask when determining antibacterial activity, thus a small spectrum of Gram-positive and one Gram-negative cell line was used. In addition, some resistant and biofilm-forming cell lines were included. Finally, some cytotoxicity data was recorded.
Altogether, this is a relevant first set of investigations for a new series of compounds, however, some of the conclusions of the benefits of these findings are a bit overinterpreted.
With this in mind, some questions are stated and some minor adjustments to the text are suggested as indicated below. Thereafter the paper can be recommended for publication in Molecules, to the benefit of other researchers in the field.
· The biological activities of many linezolid derivatives are published, a further discussion and references to close analogues would be valuable in the introduction.
· The active compounds 7i-7l are not active in linezolid-resistant cell lines, however, you see activity in LRSA and LRSP cell lines under biofilm conditions, is there an explanation for this? Moreover, linezolid should be included in this table.
· Even though there is cross-resistance with linezolid the computational studies indicate more than predict the same mode of action for these compounds.
· The cytotoxicity assay gave a decent selectivity, however, the statement that this “was a great space for medication” is a bit overstated. A great deal of additional investigations is needed to have a good lead compound for further development, e.g. hemolysis determination, a broader spectrum of bacteria (MIC90 etc.) Bacteriostatic or bactericidal action to find benefits over linezolid and other antibiotics, and of course in vivo assessment etc..
· Along the same line, a bit overstated in the conclusion “Based on these experimental results, these compounds merit lead compound status for the further development of antibacterial drugs".
Only minor linguistic corrections may be needed, the paper is generally well-written using simple and straightforward language.
Author Response
- The biological activities of many linezolid derivatives are published, a further discussion and references to close analogues would be valuable in the introduction.
Reply:The references about linezolid derivatives have been cited in the introduction, such as 17、18 and 19.
- The active compounds 7i-7l are not active in linezolid-resistant cell lines, however, you see activity in LRSA and LRSP cell lines under biofilm conditions, is there an explanation for this? Moreover, linezolid should be included in this table
Reply:We have supplemented the results of linezolid against 4 drug-resistant strains in the table 4. And what we found is that the compounds have no significant bacteriostatic effect on linezolid-resistant bacteria, indicating that its bacteriostatic mode may be similar to linezolid. But it still has a good ability to destroy the biofilm which is different from linezolid, indicating that the compounds may destroy the biofilm of bacteria in other ways. And its in-depth effect on the antibiofilm mechanism will be our next research focus.
3.The cytotoxicity assay gave a decent selectivity, however, the statement that this “was a great space for medication” is a bit overstated. A great deal of additional investigations is needed to have a good lead compound for further development, e.g. hemolysis determination, a broader spectrum of bacteria (MIC90 etc.) Bacteriostatic or bactericidal action to find benefits over linezolid and other antibiotics, and of course in vivo assessment etc..
Reply:The “lead compound” in the article is indeed inappropriate and has been modified. The current work is to further optimize the vitro antibacterial activity of the compound on the basis of previous research, and test its antibacterial effect and antibiofilm effect on drug-resistant bacteria, preliminary evaluation of the antibacterial effect of the compound in vitro, and its further extensive biological testing work and in vivo efficacy evaluation work is underway.
4.Even though there is cross-resistance with linezolid the computational studies indicate morethan predict the same mode of action for these compounds.
Reply:We very agree the opinion “Even though there is cross-resistance with linezolid the computational studies indicate more than predict the same mode of action for these compounds.”. If the compounds are useful to the linezolid-resistant drug, this will indicate the compounds have not the similar mechanism with linezolid. Instead, which indicates the compounds have the similar mechanism with linezolid. According to results of the experiments, we could discover compounds 7i-l are not useful to the linezolid-resistant drugs, we predict the compounds 7i-l have the similar mechanism with linezolid. Through computer prediction, the combination with the target is preliminarily evaluated, and the combination can also guide our subsequent structural optimization work.
5.Along the same line, a bit overstated in the conclusion “Based on these experimental results, these compounds merit lead compound status for the further development of antibacterial drugs".
“Based on these experimental results, these compounds merit lead compound status for the further development of antibacterial drugs” is indeed inappropriate and has been modified.

Round 2
Reviewer 1 Report
The authors have made some changes to the manuscript. However, now I have serious concerns on the data obtained. According to the authors’ reply (“There is a little organic solvent residue in the compounds, we have marked the corresponding solvents peak in the spectrum, and we can determine that the compounds used in the experiment are pure by high-performance liquid chromatography.”), biological evaluation was carried out on the samples that contained significant amounts of solvents and other impurities, which makes these results unreliable. Thus, I recommend this manuscript to be rejected.
In the case, if the editor decides the manuscript can be accepted, I recommend one more change to the manuscript:
1. The 3.1. Materials and Methods section should contain data on the HPLC system used to check the purity of the compounds.
Author Response
The HPLC assays of compounds purity have been added to the materials and methods.